



# Seasonal weather regimes in the North Atlantic region: towards new seasonality?

Florentin Breton[1], Mathieu Vrac[1], Pascal Yiou[1], Pradeebane Vaittinada Ayar[2], and Aglaé Jézéquel[3]

[1]Laboratoire des Sciences du Climat et de l'Environnement, UMR8212 CEA – CNRS – UVSQ, Université Paris-Saclay and IPSL, Orme des Merisiers, Gif-sur-Yvette, France
[2]Institut National de la Recherche Scientifique | INRS · Eau Terre Environnement Centre, Québec, Canada
[3]LMD/IPSL, Ecole Normale Superieure, PSL research University, Paris, France

**Correspondence:** Florentin Breton (florentin.breton@lsce.ipsl.fr)

**Abstract.** European climate variability is shaped by atmospheric dynamics and local physical processes over the North Atlantic region. As both have strong seasonal features, a better insight of their future seasonality is essential to anticipate changes in weather conditions for human and natural systems. We explore the weather seasonality of the North Atlantic over 1979-2017 and 1979-2100 by using seasonal weather regimes (SWRs) defined by clustering year-round daily fields of geopotential height

at 500 hPa (Z500) from the ERA-Interim reanalysis and 12 climate models of the Coupled Model Intercomparison Project fifth phase (CMIP5). The spatial and temporal variability of SWR structures is investigated, as well as associated patterns of surface air temperature. Although the climate models have biases, they reproduce structures and evolutions of SWRs similar to the reanalysis over 1979-2017: decreasing frequency of winter conditions (starting later and ending earlier in the year) and increasing frequency of summer conditions (starting earlier and ending later). These changes are stronger over 1979-2100 than

over 1979-2017, associated with a large increase of North Atlantic seasonal mean Z500 and temperature. When using more SWRs (i.e. more freedom in the definition of seasonality), the changes over 1979-2100 correspond to a long-term swap between SWRs, resulting in similar structures (seasonal cycle and weather patterns) with respect to the evolution of the seasonal cycle of Z500 and temperature. To understand whether the evolution of the SWRs is linked to uniform Z500 increase (i.e. uniform warming), or to changes in Z500 spatial patterns (i.e. changes in circulation patterns), we remove the calendar trend in the

Z500 regional average to define SWRs based on detrended data (d-SWRs). The temporal properties of d-SWRs appear almost constant, whereas their spatial patterns change. This indicates that the calendar Z500 regional trend drives the evolution of the SWRs and that the changing spatial patterns in d-SWRs account for the heterogeneity of this trend. Our study suggests that historical winter conditions will continue to decrease in the future while historical summer conditions continue to increase. It also suggests that according to an increasing seasonal mean, the seasonality of weather conditions would not change in a major

way.



# 1 Introduction

Are seasons changing? If so, are those changes due to climate change or to natural variability? It appears that the answers might strongly depend on the definition of season. Indeed, many investigations of seasonality have been carried out based on different definitions of the seasons (e.g. Jia et al. (2019), Stocker et al. (2013)). These investigations found changes both regarding variables in climatological seasons (e.g. decreasing winter and spring frost, decreasing summer Arctic sea ice) and regarding the seasonality of variables themselves (e.g. longer growing season, longer fire weather season). The meteorological seasons are a prominent feature of climate variability, experienced by natural and human systems through the seasonality of surface weather conditions. In Europe, these conditions mostly result from the combined effects of large-scale circulation dynamics over the North Atlantic and local-scale processes that reduce or amplify dynamic effects (Cattiaux (2010)). In this paper, we will define and investigate seasonality based on synoptic atmospheric circulation.

North Atlantic atmospheric patterns are the results of two systems operating at different scales: "low-frequency" quasi-static structures (such as the Icelandic Low and the Azores High) and "high-frequency" propagating synoptic waves (rupture of geostrophic equilibrium) or storms (Cattiaux (2010), Hurrell et al. (2003)) associated to the unstable nature of the westerly jet stream (Cassou (2008)). Due to these low-frequency and high-frequency components, the atmospheric dynamic variability of the North Atlantic is organized into preferential configurations (i.e. modes of variability) despite having a stochastic nature. One way to study these modes of variability or weather patterns is through weather regimes (WRs), defined as recurring atmospheric patterns (e.g. Vrac and Yiou (2010)). Since their first use in the middle of the XXth century in meteorology (Lamb (1950), Rex (1950)), WRs have been reintroduced in the beginning of 1980s (e.g. Reinhold and Pierrehumbert (1982)) and largely used to better understand the variability of atmospheric dynamics (e.g. Vautard (1990)), but also to evaluate climate models (Sanchez-Gomez et al. (2009), Díaz-Esteban et al. (2020)).

Both climate dynamics and weather extremes have strong seasonal features. For instance atmospheric blocking conditions facilitate cold spells in winter (Sillmann et al. (2011)) and heatwaves in summer (Schaller et al. (2018)). Furthermore, the seasonality of atmospheric dynamics has changed in the last decades with a lengthening of the period with summer conditions, starting earlier and ending later, and a shortening of the period with winter conditions (Vrac et al. (2014)). When defining seasons based on the relationship between sea level pressure and surface air temperature, Cassou and Cattiaux (2016) found that the earliness of summer conditions should continue to increase in the future while no trend is found for winter conditions. One limitation is that these results strongly depend on the definition of the seasons and on the good representation of these seasons in the climate models. More generally, we have limited confidence in the representation of atmospheric circulation in models, and the confidence in the understanding of dynamic aspects of climate change is lower than for thermodynamic aspects (Shepherd (2014)). Therefore, it is essential to evaluate how models reproduce seasonality over a historical time period. This is a necessary step prior to investigating future seasonality changes based on a non-stationary definition of seasons.

In the present paper, we investigate synoptic climatological seasonality in the North Atlantic region through the use of seasonal WRs (SWRs, Vrac et al. (2014)) that are defined by the probabilistic clustering of daily conditions of atmospheric circulation over a given time period without a priori separation of seasons. The evolution of weather seasonality is then investigated



through the variability of SWRs (structures, trends). SWRs were developed to investigate non-stationary weather seasonality through their ability to represent the evolution of atmospheric circulation modes (Vrac et al. (2007)) with season-like behavior (Vrac et al. (2014)). The large-scale increase in geopotential height at 500 hPa (Z500) due to human influence is expected to drive SWR evolutions (Christidis and Stott (2015)). To test if SWR evolutions are driven by this increase, or by changes in spatial patterns of circulation, we also look at SWRs obtained from detrended geopotential height data. This investigation allows us to remove the effects of the large-scale average Z500 increase and therefore to disentangle the potential causes of the temporal and spatial SWR evolutions. The scientific objectives of the present study are to answer the following questions:

- How are climate models able to represent past seasonal variability over 1979-2017 with respect to reanalyses?

- What is the temporal and spatial evolution of seasonal structures over 1979-2100?

- What are the causes of seasonal evolutions?

The paper is organized as follows: Section 2 describes the reanalysis and climate model data used in this study, as well as the clustering method to define seasonal weather regimes; Section 3 displays the results; and in Section 4, we discuss the findings and conclude.

## 2 Methods

### 2.1 Data and preprocessing

We use daily fields of geopotential height at 500 hPa (Z500) as a proxy of atmospheric circulation from the ERA-Interim (hereafter ERAI) reanalysis dataset (0.75° x 0.75° spatial resolution; Dee et al. (2011)) and simulations from 12 climate models of the Coupled Model Intercomparison Project fifth phase (CMIP5; Taylor et al. (2012)) over the North Atlantic region (22.5 to 70.5°N, 77.25°W to 37.5°E) from 1979 to 2017, and then from 1979 to 2100 (the datasets are briefly described in Table 1). Daily surface air temperatures (TAS) from the same datasets are also extracted to study temperature features of SWRs.
Raw year-round data is used rather than seasonal (e.g. summer or winter) data or deseasonalized anomalies to capture both the year-round seasonal cycle and any long-term trend. In order to make the analyses and comparisons easier, all datasets are first given the same format. Calendars are standardized to 365 days per year ignoring bisextile years except for the Hadley Center simulations (year of 360 days). Historical experiment runs from climate models over 1979-2005 are concatenated to RCP8.5 experiment runs over 2006-2100 (respectively 1981-2005 and 2006-2099 for the Hadley Center model). The spatial grids of data from climate model simulations are bilinearly interpolated to the ERAI grid.
A principal component analysis (PCA) is applied to the regridded Z500 fields in order to reduce the dimension of the data while keeping most of the variability and seasonality. The raw Z500 data are scaled by the square root of the cosine of the latitude to give equivalent weight to all grid cells when performing the PCA (as in e.g. Cassou (2008)). Only the first principal component (PC1) is kept and used for clustering because it captures between about 49% and 60% of the variance and between about 95% and 99% of the seasonal cycle (spectral power at $1/365$ of frequency; $1/360$ for Hadley Center) over 1979-2017





for ERAI (similar to Vrac et al. (2014) on another reanalysis) and all climate models (not shown). Including more PCs in the analysis provided similar results (not shown) but brought more noise (more variance but only little more seasonality).

## 2.2 Definition of seasonal weather regimes

We use the Expectation-Maximization (EM) algorithm (Dempster et al. (1977)) based on a Gaussian mixture model (GMM; Peel and McLachlan (2000)) to cluster probabilistically the 14235 days (13320 for Hadley Center) of the 1979-2017 period into Seasonal Weather Regimes (SWRs). The EM algorithm estimates a multivariate probability density function (pdf) $f$ of the data (here, daily PC1 values) as a weighted sum of $K$ Gaussian pdfs $f_k$ ($k = 1, \ldots, K$) (Pearson (1894)):

$$f(x) = \sum_{k=1}^{K} \pi_k f_k(x; \alpha_k) \qquad (1)$$

where $\alpha_k$ contains the parameters (means $\mu_k$ and covariance matrix $\Sigma_k$) of $f_k$ and $\pi_k$ is the mixture ratio corresponding to the prior probability that $x$ (i.e. PC1 value) belongs to $f_k$. The parameters $\alpha_k$ and $\pi_k$ ($k = 1, \ldots, K$) of the GMM are unknown and must be estimated (cf. Appendix). Finally, each cluster $C_k$ of days is defined based on the Gaussian pdfs, according to the principle of posterior maximum:

$$C_k = \{x; \pi_k f_x(x; \alpha_k) \geqslant \pi_j f_j(x; \alpha_j), \qquad \forall j = 1, \ldots K\} \qquad (2)$$

In other words, each day is assigned to the cluster for which the probability of belonging is maximum, and the obtained clusters are SWRs which correspond to a classification of the daily data. The freedom of EM in the definition of the SWRs strongly depends on the number $K$ of clusters and on the constraints applied to the covariance matrices (constraining the geometry of the clusters, cf. Appendix). We tried different values for $K$ (from $K = 1$ to $K = 15$) and evaluated them through the Bayesian Information Criterion (BIC; Schwarz et al. (1978)). Optimizing the BIC achieves a compromise between overfitting 105 the observations with the model and the complexity of the model (cf. Appendix). Four SWRs (hereafter SWR4) correspond both to a plateau of BIC (not shown) and to the traditional (astronomical) number of seasons. The GMM with the best BIC is selected. Different clustering methods can lead to different results (e.g., Philipp et al. (2010)) so we tested the sensitivity of the SWR results to using k-means instead of EM clustering, which brought very similar results (not shown).

## 2.3 Seasonal weather regimes based on detrended data

We use the same method as previously but after first removing the large-scale spatial Z500 trend from the original data while keeping the seasons and spatial patterns. This detrending is done by estimating and removing the calendar trend in the Z500 regional average, and adding the estimated seasonal cycle of 2017 (details in Appendix). The detrended SWRs (d-SWRs) are then obtained by applying the same method as before, of principal component analysis followed by clustering, to the detrended data. We heuristically chose to detrend TAS for this part of our analysis, similarly to Z500. We emphasize that the detrending 115 removes the regional (large-scale) calendar trend so that the resulting local trend of the detrended data is the residual of the regional trend. A negative residual trend at a given gridpoint means that its Z500 values are increasing less than the regional





average, or even are decreasing, whereas a positive residual trend means that the local trend is higher than that of the regional average.

## 3 Results

The first part of the results focuses on the SWRs in ERAI and in the climate models over 1979-2017 to assess how the models perform with respect to the reanalyses. The second part examines SWRs in climate models over 1979-2100 to detect evolutions in the temporal and spatial structures. The third part explores the possible causes for the evolution of SWRs over 1979-2100, such as uniform large-scale Z500 increase, or changes in Z500 spatial patterns.

### 3.1 Evaluation of past seasonal weather regimes in climate models (1979-2017)

We start by looking at the annual cycle of the regimes monthly frequencies over 1979-2017, shown in Figure 1. Climate models reproduce a seasonal cycle of SWRs similar to ERAI, with regime 1 (hereafter R1) representing a winter-like season, R4 a summer-like season, and R2 and R3 transitional seasons (R2 around winter and R3 around summer). The composite maps associated with each regime are shown in Figure 2. For climate models, each regime composite map is determined individually (i.e. average map) and the multimodel composite is calculated as the mean of the distribution of the twelve composites. The

spatial patterns of the four average regimes found in the models are very similar to those from ERAI. They are also visually similar to the usual North-Atlantic weather regimes from the literature (e.g. Cassou (2008), Yiou and Nogaj (2004)).

The first regime (R1) corresponds to the positive phase of the North Atlantic Oscillation (NAO+) and the second (R2) to its negative phase (NAO-; Hurrell et al. (2003)). The third and fourth regimes (R3 and R4) respectively represent the Atlantic Ridge (AR) and Scandinavian Blocking (SB) atmospheric conditions, resembling the weather patterns from Yiou and Nogaj

(2004), and Vrac et al. (2014). However, note that the temporal patterns of our SWRs are based on full years (like Vrac et al. (2014)), unlike the literature considering weather patterns in winter (Cassou (2008), Yiou and Nogaj (2004)) or in summer (e.g. Guemas et al. (2010)). Thus, if our seasonal weather regimes resemble the usual regimes, they present differences in their definition and then in their properties.

In general, the climate models reproduce atmospheric weather patterns that are very similar to ERAI, but individual models

are less successful (see Supplementary Fig. 1-4). For example, the weather patterns associated with R1 and R3 in MIROC5 are very different from other climate models and ERAI, despite happening at the same period in the year (Fig. 1). All other climate models show R1 patterns similar to ERAI albeit with diverse intensities. In the case of R2 and R4, ERAI and all climate models agree on the circulation pattern but differ in intensity. The representation of R3 in climate models is slightly biased by comparison with ERAI because of inaccurate flow intensity and location of pressure centers. The variability between climate

models, represented here by the standard deviation over the 12 values (one per regime composite of climate model), appears larger (Fig. 2) for regimes deviating from the seasonal mean atmospheric circulation (e.g. R1 with intense colors) than for regimes following it (e.g. R3 with pale colors).

After looking at the seasonal structure of the regimes, we investigate if and how the temporal organisation of these regimes





changes during 1979-2017 through (i) the regime monthly frequencies, (ii) the first (start) and last (end) days of regime

occurrence, and (iii) the regime persistence (i.e. average number of consecutive days). Most changes in ERAI and in the

average of the models are similar (not shown) to the results from Vrac et al. (2014) on NCEP reanalyses: R1 (i.e. winter

conditions) decreasing in frequency, starting slightly later, ending slightly earlier, and being less persistent, and the opposite

for R4 (i.e., summer conditions). The spatial patterns of TAS associated with the regimes are also similar to Vrac et al. (2014)

(Supplementary Fig. 5).

## 155    3.2    Future changes in seasonal weather regimes (1979-2100)

We now use the same method as before to define SWRs but based on the full simulation datasets over 1979-2100 to detect

potential future changes. The first approach is to use four regimes (SWR4). Between the first three decades (1979-2008) and

the last three decades (2071-2100) of the period, R1 (NAO+) occurs less often but is more intense for both Z500 and TAS

(Supplementary Fig. 6-7). The opposite happens for R4 (SB) that occurs more often with less intense patterns, i.e. becoming

closer to the seasonal mean. R2 (NAO-) occurs more often but is less intense, while R3 (AR) occurs slightly less often but is

more intense. Note that these patterns are relative to the seasonal mean, which increases substantially over the North Atlantic

between the first and last three decades (averaging about +90 hPa for Z500 and +4°C for TAS; not shown).

A shift happens in the annual cycle of SWR4 over 1979-2100 with R4 growing, R2 and R3 moving towards the winter period

of the year, and R1 shrinking in time (Supplementary Figure 8). GFDL-CM3 stands out from the other GCMs by showing

the emergence in the future of a new summer regime that did not exist in the past. This is consistent with higher increases of

Z500 and TAS in this climate model by comparison to other models (not shown). Monthly frequencies show R2 taking the

place of R1 in the December, January and February months (hereafter DJF) starting from the middle of the 21st century, and

R4 taking the place of R3 in June around 2025 (not shown). Although the average between models shows a clear direction of

SWR evolution, the timing of this evolution differs up to a few decades between individual models. In consistence with the

seasonal shift of regimes, the average of climate models between 1979 and 2100 shows R1 starting about one month and a half

later while ending about two months earlier, and persisting less, whereas R4 starts about one month earlier while ending about

one month and a half later, and persists more (Supplementary Fig. 9-10).

Over 1979-2100, SWR spatial trends of Z500 and TAS are in agreement between GCMs (Supplementary Fig. 11-12) and

are more robust than over 1979-2017. These maps of linear trends are obtained by calculating the linear regression of the

evolution of the variable (raw values) by gridcell, grey areas correspond to trends that are not significant (p-value > 0.05).

Both regression values and p-values are calculated individually by climate model, and then averaged over the twelve values.

However, these SWR spatial trends show different spatial evolutions between Z500 and TAS within regimes, hence partially

decoupled evolutions of atmospheric dynamics and surface temperature.

Even if using four regimes allows us to explore the future with a traditional number of seasons, the low number of clusters limits

the freedom of the clustering to allow the appearance or disappearance of significant structures. Therefore, we applied a second

approach to overcome this limit. We tested different numbers of regimes and chose seven regimes as a showcase because it

illustrates the clearest transitions between the disappearance of past structures and appearance of future (new) structures.



With seven regimes (SWR7), the patterns of atmospheric circulation are very similar to those of surface temperatures in both past (1979-2008) and future (2071-2100) (Figures 3-4). Regime patterns seem to follow the seasonal cycle (pale colors) except

R1, R2 and R7. Past (1979-2008) R7 corresponds to rare and very intense conditions of Scandinavian Blocking associated with summer heatwaves over Northern continents. Future (2071-2100) R1 corresponds to rare and very intense NAO+ conditions associated with cold spells over Northeastern America, Greenland and Scandinavia.

Overall, we observe a shift in the spatial patterns (Z500 and TAS) of the regimes (Figures 3-4) with past R1 patterns becoming future R2 patterns, past R2 patterns becoming future R3 patterns, and so on until R6, while R1 pattern becomes seasonally

more extreme (rarer and more intense pattern) and R7 pattern becomes seasonally more normal (more frequent and less intense pattern). We calculated the average seasonal cycle of the seven regimes in a similar way to Fig. 1 but over the first three decades (1979-2008) and the last three decades (2071-2100), shown in Figure 5. R7 is a new summer regime almost absent in the past period (1979-2008) that replaces R6 and "pushes" all the other regimes towards the winter calendar days while R1 (past or old winter regime) collapses until almost disappearance. This shift in the seasonal cycle of the regimes between past and future

appears very consistent with the shift in the regime spatial patterns.

The timing of these changes in regime occurrence during the year can be investigated through the monthly frequencies of the regimes over 1979-2100 (winter months in Figure 6 and summer months in Figure 7). Figure 6 shows the collapse of R1 happening throughout the 21st century. R2 takes the place of R1 in the beginning of the 21st century, and becomes replaced by R3 at the end of the 21st century. Symmetrically, R6 is replaced by R7 during the second half of the 21st century (Figure

7). The evolution of the starting and ending dates as well as persistence of R1 and R7 are very consistent with the evolution of their seasonal cycle and monthly occurrence (Supplementary Fig. 13-14).

All regimes except R7 show a similar pattern of Z500 change over the region: increase in the Southern part and decrease in the Northern part, whereas R7 shows widespread increase that is stronger in the South and not robust between climate models in the North of the region (Supplementary Fig. 15). Interestingly, these changes in circulation patterns seem to be opposite to

the expected effects from Arctic amplification, such as amplified warming and geopotential height increase over circulation dynamics that are linked to midlatitude weather (Barnes and Polvani (2015), Cohen et al. (2014), Overland et al. (2015)). The strongest warming over the region is observed in R1 and R7, whereas R3 to R6 show (unexpected) cooling over the continents (Supplementary Fig. 16). The origin of this cooling is investigated later in the discussion of the paper (Section 4.3). The appearance and disappearance of regimes observed in SWR7 over 1979-2100 is absent from the 1979-2017 period where we

tested with four up to seven regimes.

### 3.3  Seasonal weather regimes based on detrended data (1979-2100)

The increasing trend of Z500 over the North Atlantic region, mainly due to human influence (Christidis and Stott (2015)), is expected to be driving the evolution of the SWRs but changes in spatial patterns could also play a role. To investigate this, we use SWRs based on detrended data (d-SWRs) and focus on the average d-SWRs of climate models. This detrending corresponds

to removing the calendar (by day in the year) trend of the regional average Z500 (or TAS, see Methods 2.3). By comparison to SWRs, the temporal structures of d-SWRs over 1979-2100 appear almost stationary and remain very similar to those of ERAI





(Figure 8). However, spatial structures of d-SWRs present some minor variability for Z500 (Figure 9) but major changes for TAS in which case future patterns are almost symmetrically opposite to past patterns (Figure 10). This small evolution of Z500 spatial patterns in d-SWRs can be explained by spatial trends that are either not significant in individual climate models or in disagreement between climate models, as shown by large greyed areas in Supplementary Fig. 17. However, most of TAS spatial trends in d-SWRs are robust and show warming over continents and cooling over oceans (Supplementary Fig. 18). This warming contrast can be explained because of the higher heat capacity and evaporative cooling potential of ocean surface than land surface, and ocean mixing (e.g. Dai (2016)). These trends also show Arctic amplification (i.e. warming stronger at the pole than at lower latitudes), especially in winter (R1 to R3).

To further understand the roles of the large-scale increases in Z500 and TAS (hereafter LGI), and of the seasonal shift of regimes (driven by the large-scale Z500 increase) in the changes of Z500 and TAS patterns within SWRs and d-SWRs, we examine the regime spatial trends with LGI but without the seasonal shift (Supplementary Fig. 19-20), and with the seasonal shift but without LGI (Supplementary Fig. 21-22). The contribution of the large-scale increase of Z500 and TAS is investigated through the regime composite maps calculated based on Z500 (or TAS), conditionally to the clusters defined on detrended Z500 (i.e. clusters with almost constant temporal structures). The contribution of the seasonal shift of regimes is investigated through the regime composite maps calculated on detrended Z500 (or detrended TAS), conditionally to the clusters defined on Z500. The contribution of LGI corresponds only to widespread increasing Z500 and TAS in all regimes whereas the shift of SWRs towards winter corresponds to widespread decreasing Z500 and TAS in most regimes (except R1, R2, and unconditionally to regimes). The two opposing effects of LGI and the seasonal shift can explain the existence of decreasing spatial trends of Z500 and TAS observed earlier within SWRs.

## 4  Conclusive discussions

We used seasonal weather patterns (Vrac et al. (2014)) by clustering Z500 from the ERAI reanalysis and 12 CMIP5 climate models to study past (1979-2017) and future (1979-2100) seasonal structures of mid-troposphere atmospheric dynamics (Z500) and air surface temperature (TAS) over the North Atlantic region and their evolutions in time.

### 4.1  Ability of climate models to represent past seasonal variability

Despite biases in the climate model representation of Z500 and TAS seasonal variability, the general evolution of the four seasonal weather regimes (SWR4) over 1979-2017 was consistent between models and ERAI: decreasing frequencies of historical winter conditions and increasing frequencies of historical summer conditions of atmospheric dynamics. Most of the results agree with the findings of Vrac et al. (2014), except that we detect a more pronounced winter evolution. This is probably because their reanalysis dataset covered only 1948-2011 while ours are over 1979-2017, which is more recent and better captures global warming (section 2.4.1.1 and Table 2.4 in Stocker et al. (2013)). The structures (spatial, temporal) and evolution (timing) of SWRs differ between climate models over 1979-2017 and even more over 1979-2100.





## 4.2 Projected evolutions of seasons

When looking at future (1979-2100) evolutions of SWRs with both four and seven regimes, the frequency of historical winter
conditions decreases while that of historical summer conditions increases and occurrences of transitional regimes move towards
the winter period. These changes are attached to large increases in the seasonal mean of Z500 and TAS over the North Atlantic.
The results for summer are consistent with those of Cassou and Cattiaux (2016) but not the results for winter, which could be
due to the very different methods used to define seasonality. Moreover, allowing for more freedom in the definition of the SWRs
by using seven regimes rather than four, we find a collapse of the regime associated to past winter conditions, corresponding
to rare cold spells at the end of the 21st century, and the growth of a new summer regime corresponding to past heatwaves that
becomes dominant in summer by the end of the 21st century.

These results suggest that past winter conditions are becoming shorter in time and past summer conditions are broadening
and intensifying. However, in our case the apparent changes in seasonality seem to correspond rather to a swap between
regimes since occurrences of past R1 are replaced by R2 in the future, past R2 are replaced by R3, and so on until R6. Note
that R1 conditions correspond to the past winter pattern that almost disappears at the end of the 21st century. Hence, for the
future projections, R1 corresponds to extreme winter (intense and rare NAO+) with respect to the "normal" future seasonality.
Therefore, this regime swap, with symmetry between spatial patterns and seasonal cycle, suggests that the seasonality of the
weather patterns does not change in a major way with respect to the evolution of the raw seasonal cycle of Z500 and TAS.

Over the last three decades (2071-2100) respectively to the first three decades (1979-2008), SWR4 had about 75% fewer days
(on average between climate models) of atmospheric conditions in the positive phase of the North Atlantic Oscillation (NAO+)
and about 10% for Atlantic Ridge (AR) configurations but about 54% more days in the negative phase of the North Atlantic
Oscillation (NAO-) and 135% more days for Scandinavian Blocking (SB) patterns. The increase of days of SB conditions
during May, June, September and October could have consequences on extreme events, such as heatwaves and dry spells
as Röthlisberger and Martius (2019) found a strong positive effect of atmospheric blocking conditions on the persistence of
simultaneously occurring hot and dry spells over Europe between May and October.

Additionally, the findings from Pfleiderer et al. (2019) that summer weather becomes more persistent in a warmer world,
although they consider summer in June-July-August, can be linked to our finding of an increase in summer regime persistence.
SWR4 over 1979-2100 also revealed a weakening of NAO- and SB weather patterns and a strengthening of NAO+ and AR
patterns at the end of the 21st century by comparison to the end of the 20th century. This future strengthening of the NAO+
atmospheric pattern in the winter period is consistent with a decrease of cold spells over Europe (Peings et al. (2013)) as they
are facilitated by SB conditions in winter (Buehler et al. (2011)). The strengthening of AR conditions and weakening of SB
conditions can be put in relation with the suggestion from Christidis and Stott (2015) that the relative Z500 increase between
polar and mid-latitude regions in the Northern Hemisphere could moderate the westerly flow over the North Atlantic and affect
the positioning of the North Atlantic jet stream, especially with a change in the sinuosity of the midlatitude atmospheric flow
(Cattiaux et al. (2016)).





### 4.3 Drivers of the evolution(s)

The appearance and disappearance of regimes over 1979-2100 do not happen in 1979-2017, probably due to the smaller scale of change in Z500 in this period by contrast to the future where the full extent of the emission scenarios kick in inside the climate model simulations. We found that spatial trends of increasing and decreasing Z500 within regimes, generally associated

respectively to TAS warming and cooling trends, are the result of two opposite processes: the large-scale increase of Z500 due to human influence, and the seasonal shift of regimes towards the winter period, where Z500 and TAS are lower than during the rest of the year. This seasonal shift explains the decreasing Z500 trends, generally associated with cooling, which are observed in several regions within SWRs and would otherwise not be possible. This explanation also covers the cooling trends reported by Vrac et al. (2014), understood here as a temporal shift of the regimes' occurrences towards the winter period with cooler

conditions rather than a seasonally-stationary cooling.

The d-SWRs results (i.e., SWRs obtained from detrended Z500) showed almost no temporal evolution between past and future, which means that the Z500 large-scale increase is the main cause for the evolution of SWRs. Christidis and Stott (2015) reported that the large-scale Z500 increase during 1979-2012 was mostly due to human forcings. So, although climate models overestimate the surface warming and Z500 increase over the past period (Christidis and Stott (2015), Jones et al. (2013)),

there might be a strong link between the human forcings and the shift in seasonality of the regimes that we detect here, since most of the evolution of the regimes disappears when we remove the calendar large-scale Z500 increase.

### 4.4 Limitations and perspectives

Even if the regimes and their evolutions in climate models in the past period are similar to those from ERAI, we note a few limitations and sources of uncertainty. The representation of the climate in ERAI and models has uncertainties and errors,

especially in atmospheric dynamics (Shepherd (2014)) and surface temperature in models (Jones et al. (2013)). Bias correction methods could lead to more realistic seasonal weather regimes but could imply other issues such as modifications of spatial and temporal structures (and trends) that could possibly generate physical inconsistencies (Vrac (2018), François (2020, in review)).

Overall, although our study highlights the value of a clustering approach for comparing (and evaluating) models as well as

seasonal structures, the apparent consistency that we find between climate models on the future evolution of seasonal dynamics seems at odds with other studies where the projected circulation response differs strongly between models (e.g. Barnes and Polvani (2015)). Indeed, clustering approaches might hide inter-model variability, or seasonal variability (depending on the number of clusters). Additional sources of uncertainty include the choice of RCP8.5 for the future emission scenarios and the choice of Z500 (i.e., mid-troposphere atmospheric circulation) rather than surface, low- or high-troposphere conditions.

Similar methods to those that we used could be applied to explore changes in weather seasonality at a more local scale by downscaling meteorological variables (e.g. humidity, wind speed, temperature) based on large-scale weather regimes (Vrac and Yiou (2010)) in order to bring more locally-relevant insights for social matters related to the weather. Understanding



recent and future changes in seasonality is important to anticipate future changes in weather conditions and the consequences for nature and society.

*Code and data availability.* The ERAI data is available at https://www.ecmwf.int/en/forecasts/datasets/reanalysis-datasets/era-interim. The CMIP5 data is available at https://esgf-node.ipsl.upmc.fr/search/cmip5-ipsl/. The computations were done using the free statistical package mclust (Scrucca et al., 2016) on the R software (www.r-project.org).

*Author contributions.* FB and MV: conceptualization of the study and investigation, writing of the original draft. PY, PVA and AJ: writing (review and editing).

*Competing interests.* The authors do not have competing interests to declare.

*Acknowledgements.* FB and MV acknowledge financial support from the CoCliServ project. MV and PY also acknowledge support from the EUPHEME project. Both CoCliServ and EUPHEME are part of ERA4CS, an ERA-NET initiated by JPI Climate and co-funded by the European Union. We also thank Soulivanh Thao and Ara Arakelian for technical assistance with the experiments and insightful discussions on the findings.



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



**Table 1.** Characteristics of data used.

| Dataset | Period | Spatial resolution (lon x lat) | Institute | Reference |
|---|---|---|---|---|
| ERA-Interim | 1979-2017 | 0.75° x 0.75° | ECMWF (Europe) | Dee et al. (2011) |
| HadGEM2-ES | 1981-2005 (historical) and 2006-2099 (RCP8.5) | 1.875° × 1.25° | MOHC (UK) | Jones et al. (2011) |
| ACCESS1-3 | 1979-2005 (historical) and 2006-2100 (RCP8.5) | 1.875° x 1.25° | CAWCR (Australia) | Collier and Uhe (2012) |
| bcc-csm1-1-m | 1979-2005 (historical) and 2006-2100 (RCP8.5) | 1.125° | BCC (China) | Wu et al. (2014) |
| CanESM2 | 1979-2005 (historical) and 2006-2100 (RCP8.5) | 2.8125° x 2.7906° | CCCma (Canada) | Chylek et al. (2011) |
| CNRM-CM5 | 1979-2005 (historical) and 2006-2100 (RCP8.5) | 1.40625° x 1.4008° | CNRM (France) | Voldoire et al. (2013) |
| GFDL-CM3 | 1979-2005 (historical) and 2006-2100 (RCP8.5) | 2.5° x 2° | GFDL (USA) | Griffies et al. (2011) |
| IPSL-CM5A-MR | 1979-2005 (historical) and 2006-2100 (RCP8.5) | 2.5° x 1.2676° | IPSL (France) | Dufresne et al. (2013) |
| IPSL-CM5B-LR | 1979-2005 (historical) and 2006-2100 (RCP8.5) | 3.75° x 1.8947° | IPSL (France) | Dufresne et al. (2013) |
| MIROC5 | 1979-2005 (historical) and 2006-2100 (RCP8.5) | 1.40625° x 1.4008° | CCSR, NIES, JAMSTEC (Japan) | Watanabe et al. (2010) |
| MPI-ESM-MR | 1979-2005 (historical) and 2006-2100 (RCP8.5) | 1.875° x 1.8653° | MPI (Germany) | Giorgetta et al. (2013) |
| MRI-ESM1 | 1979-2005 (historical) and 2006-2100 (RCP8.5) | 1.125° x 1.12148° | MRI (Japan) | Adachi et al. (2013) |
| NorESM1-M | 1979-2005 (historical) and 2006-2100 (RCP8.5) | 2.5° x 1.8947 | BCCR, NMI (Norway) | Bentsen et al. (2012) |



**Figures**





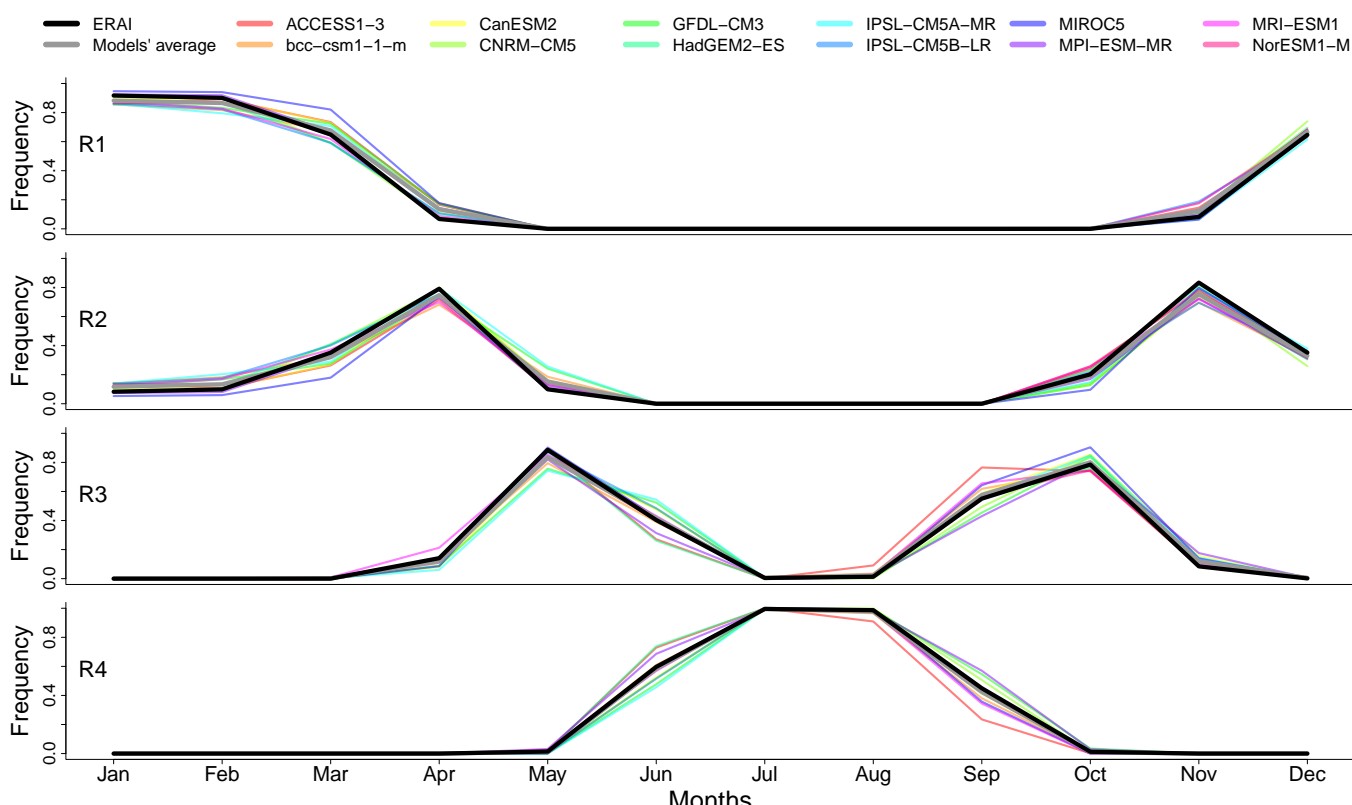

**Figure 1.** Average seasonal cycle of the frequencies of occurrences for the four regimes of ERAI and the 12 climate models, over 1979-2017.
Monthly frequencies correspond to the number of days of regime occurrence divided by the number of days in the month.



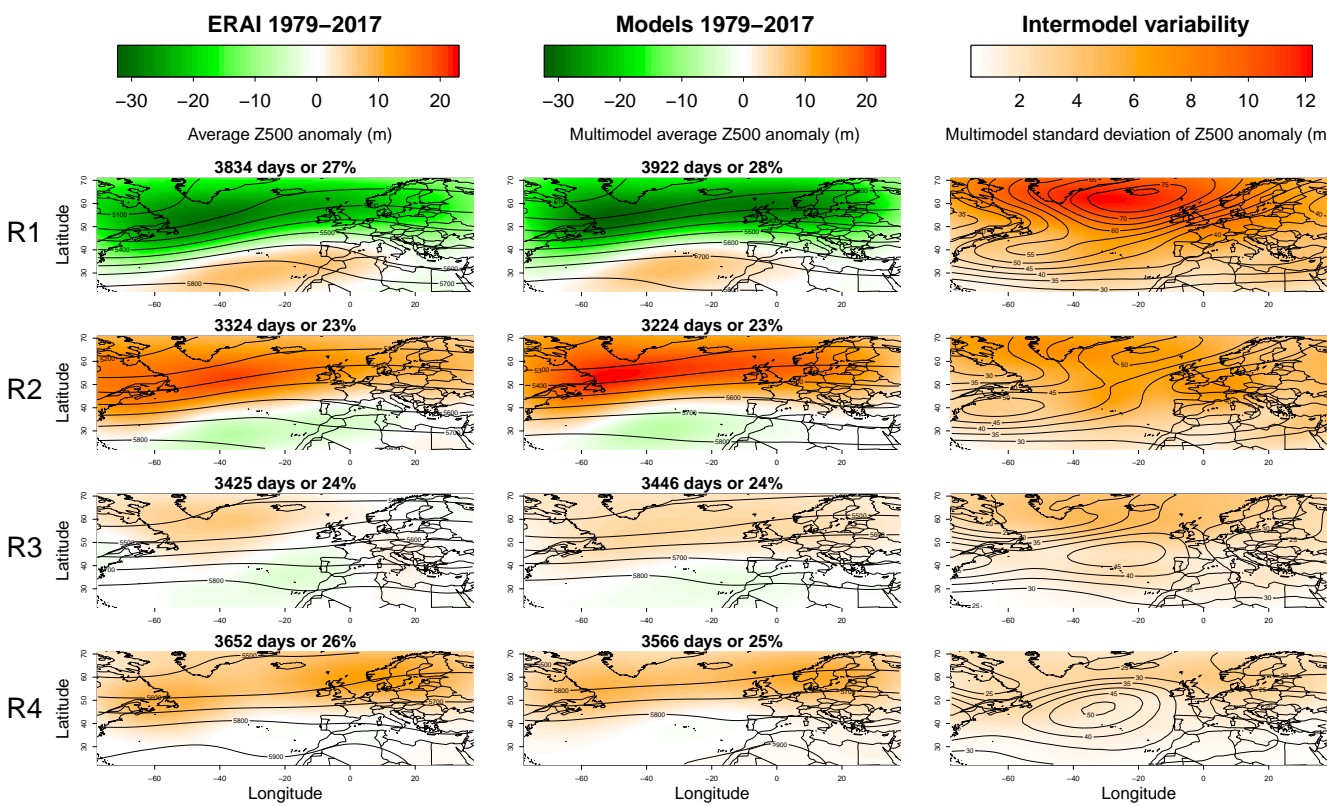

**Figure 2.** Composite maps of the four regimes (one per row) for ERAI (first column) and climate models (second column; each map shows the average of 12 composite maps; third column shows standard deviation of Z500 between the 12 composites). The maps are calculated by averaging the seasonal anomalies (shading) and raw values (contour lines) over the days belonging to the regime. Seasonal anomalies correspond to the raw values minus the average seasonal cycle. The number of days per regime is shown above each map (average of 12 values for the climate models).



**Figure 3.** Same as Fig. 2 but for the climate models (without ERAI) and seven regimes over 1979-2100. The seasonal anomalies are calculated with reference to the average seasonal cycle of the subperiod (1979-2008, 2071-2100).



**Figure 4.** As in Fig. 3 but for TAS anomalies conditionally to the regimes.



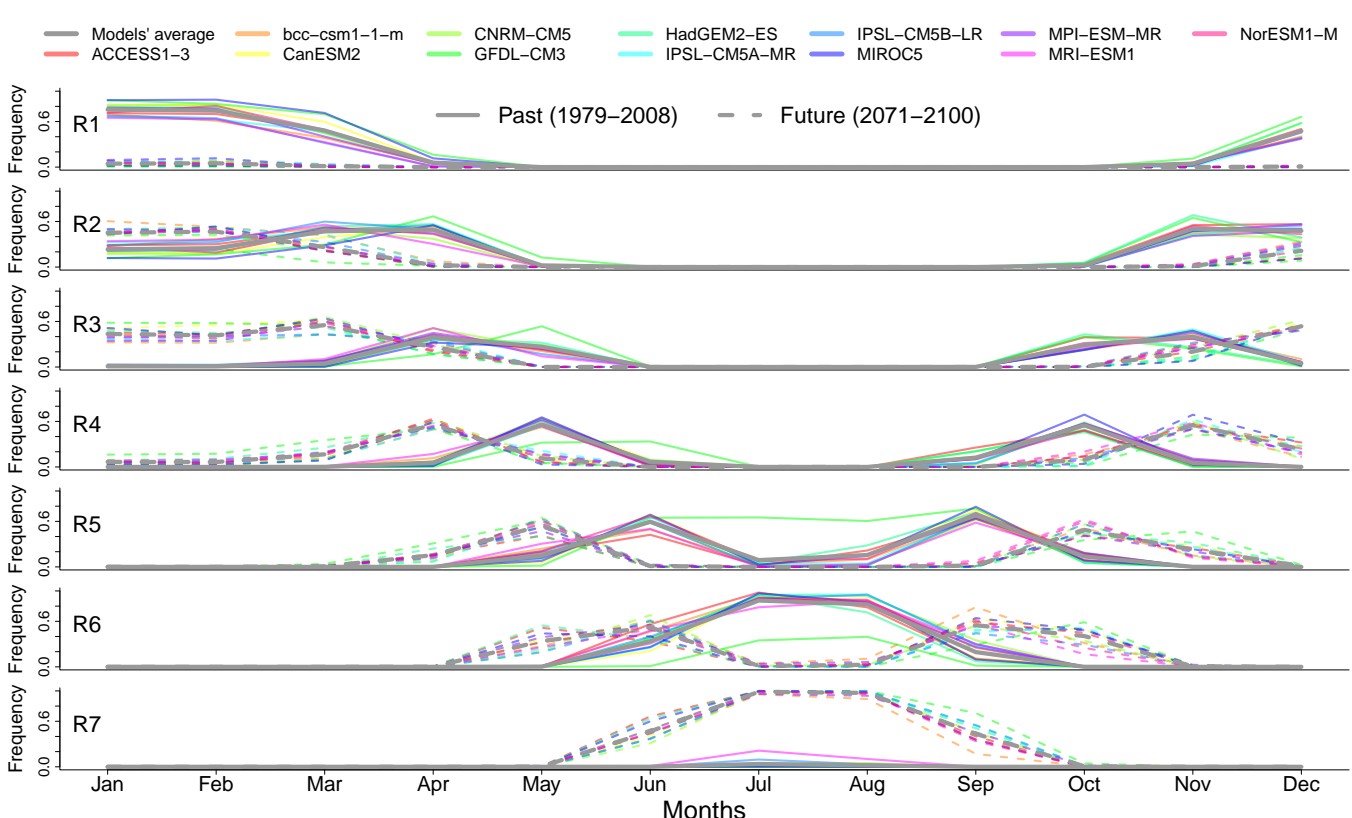

**Figure 5.** Average seasonal cycle of the seven regimes for the 12 climate models in the past (1979-2008) and future (2071-2100).





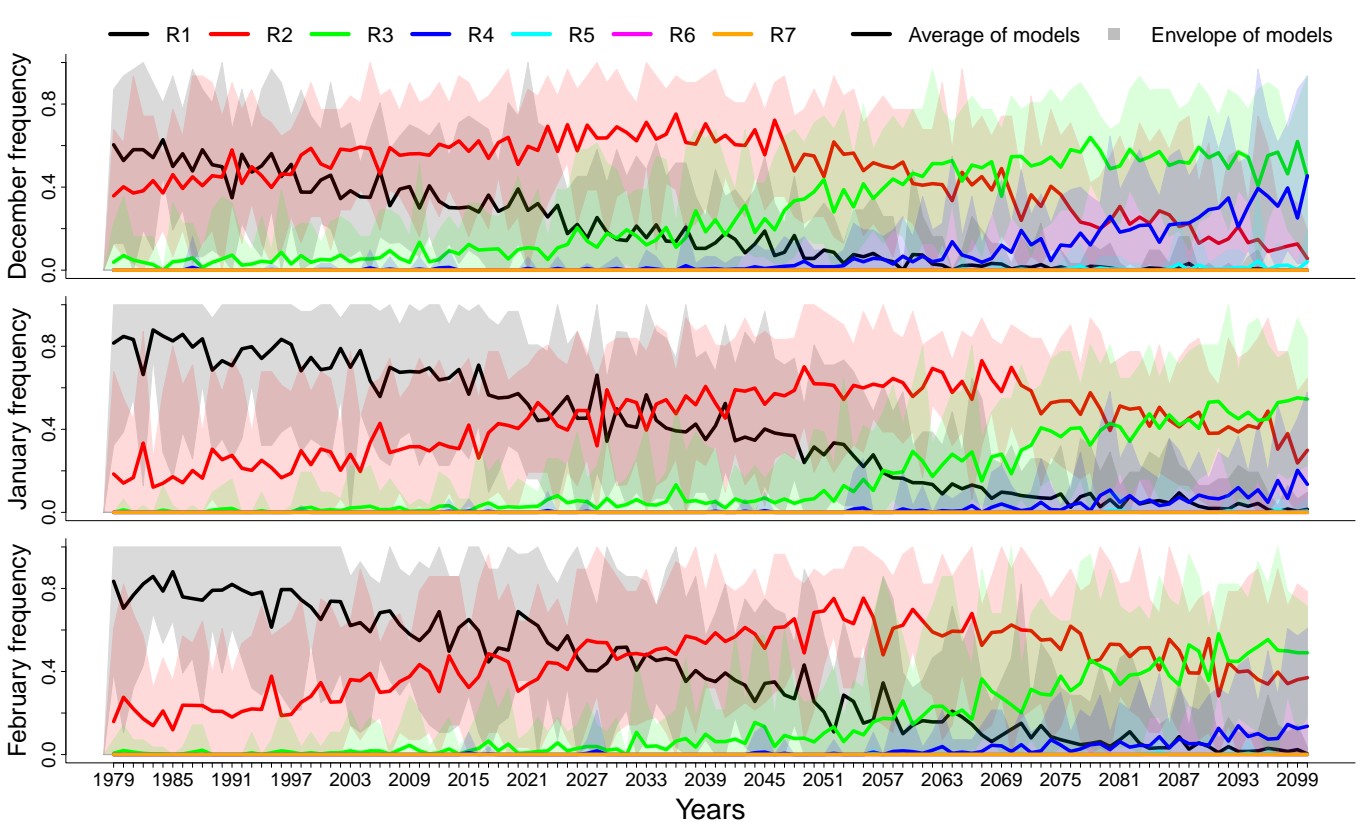

**Figure 6.** Frequency of the regimes per year in December, January and February for the 12 climate models.



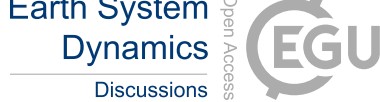

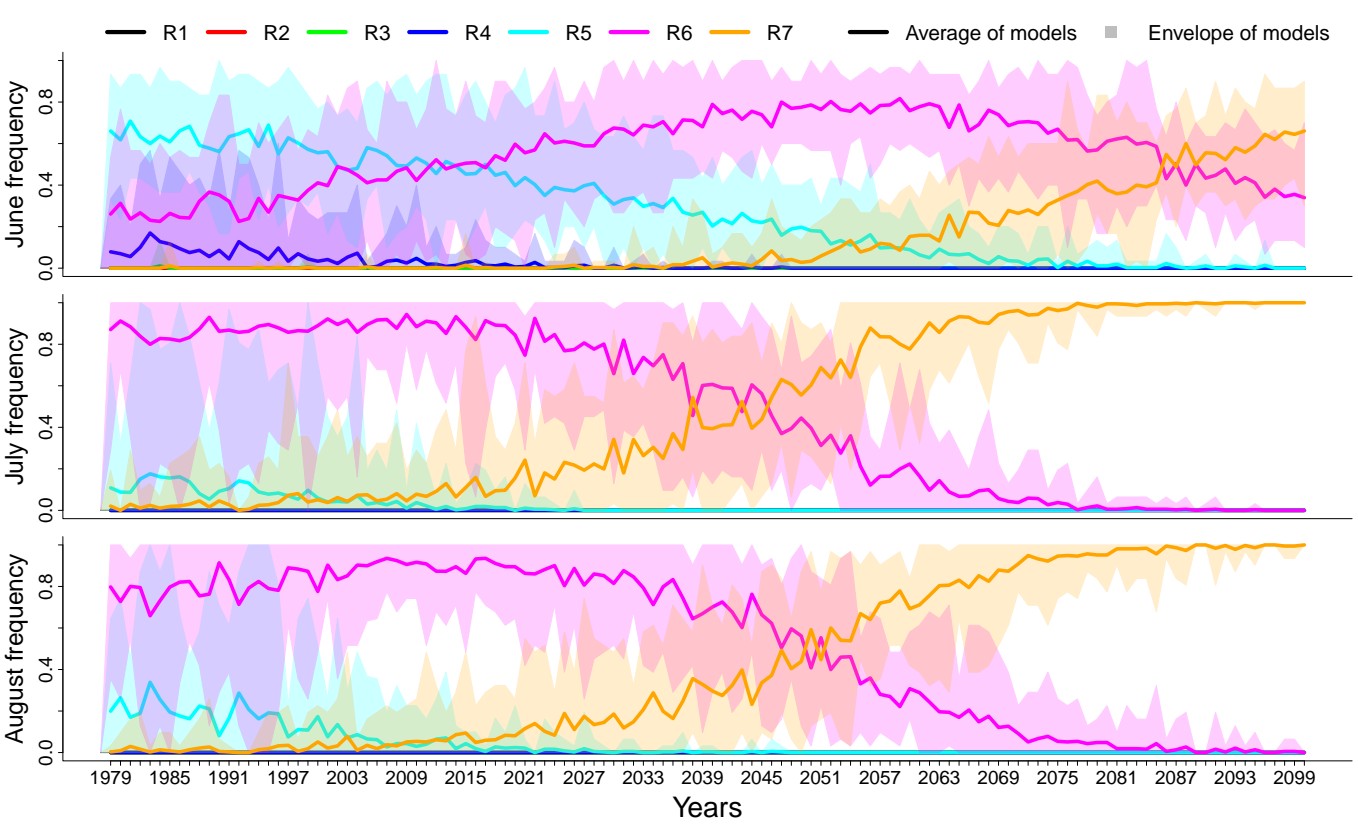

**Figure 7.** As in Fig. 6 but for June, July and August.





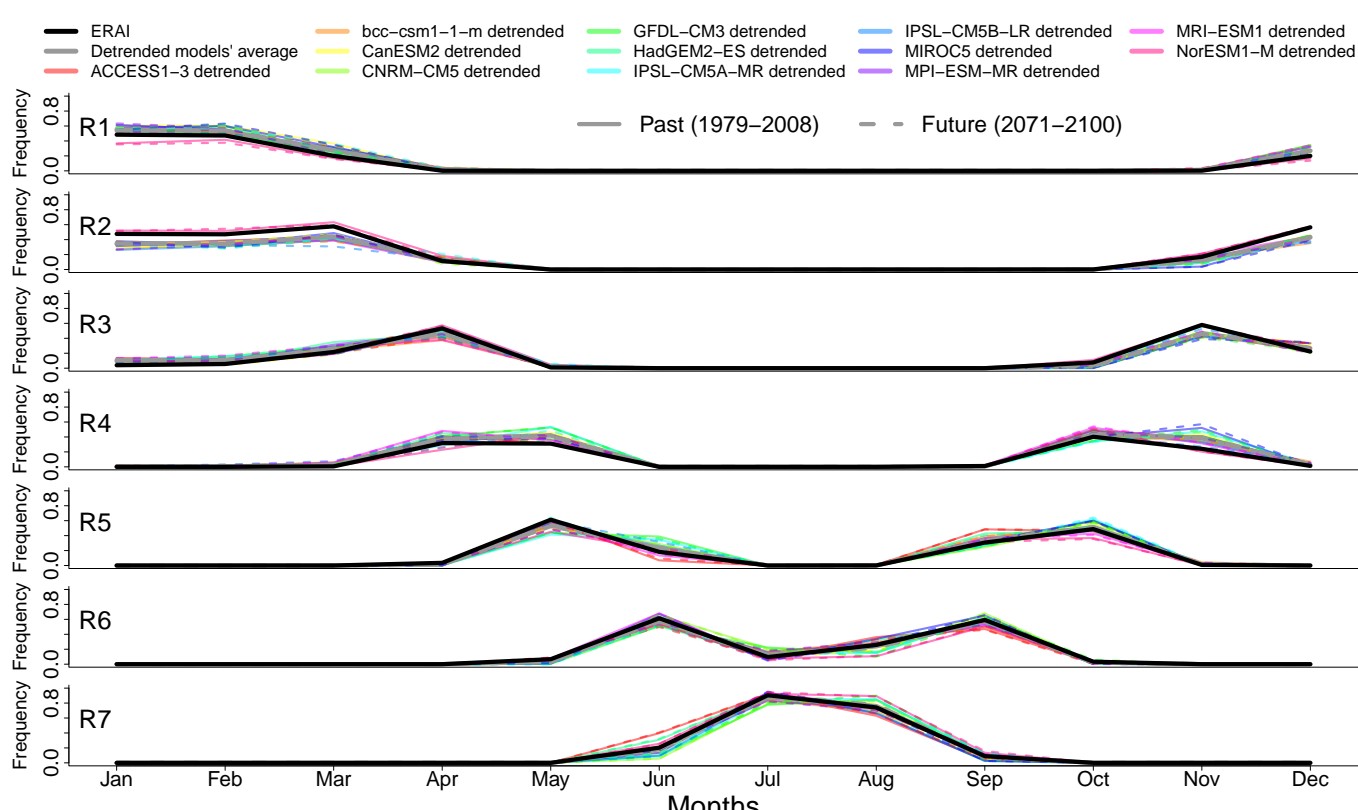

**Figure 8.** As in Fig. 5 but after detrending the data from climate models.



**Figure 9.** As in Fig. 3 but after detrending the data from climate models.



**Figure 10.** As in Fig. 4 but after detrending the data from climate models.





## Appendix A: Density estimation via Gaussian Mixture Model

Gaussian distributions are ellipsoids in space determined by the mean (location) and covariance matrix (geometric features: volume, shape, orientation). The parameters of the Gaussian Mixture Model (GMM) are the means $\mu_k$, covariance matrix $\Sigma_k$, and mixture ratio $\pi_k$, describing the $K(k = 1, \ldots, K)$ Gaussian distributions. The estimation of the GMM parameters is done iteratively in the Expectation Maximization (EM) algorithm by maximizing the likelihood that the current statistical model represents the observed data (Fraley and Raftery (2002)). Before being optimized, the GMM parameters are initialized by the result of a hierarchical model-based agglomerative clustering (multivariate), or by separation in quantiles (univariate), rather than random initialization. This approach avoids poor initial partitioning leading to the convergence of the likelihood function to a local maximum rather than a global one (e.g. Scrucca and Raftery (2015)). The principle of EM is based on the possibility to calculate $\pi$ when knowing $\alpha$ ($\mu$ and $\Sigma$) and vice-versa, thus enabling the optimization of both. After the initialization, the Expectation-step (or E-step) estimates the posterior probability $p_{ik}$ (update of $\pi_{ik}$) that the observation $x_i$ belongs to $f_k$ with the current parameter estimates (at stage $t$):

$$p_{ik}^t = \frac{\pi_k^t f_k(x_i, \alpha_k^t)}{\sum_{k=1}^{K} \pi_k^t f_k(x_i, \alpha_k^t)} \tag{A1}$$

Then, the Maximization-step (or M-step) uses the posterior probabilities to improve the estimates of GMM parameters (stage $t + 1$):

$$\pi_k^{t+1} = \frac{1}{n} \sum_{i=1}^{n} p_{ik}^t \tag{A2}$$

$$\mu_k^{t+1} = \frac{1}{n \pi_k^{t+1}} \sum_{i=1}^{n} x_i p_{ik}^t \tag{A3}$$

$$\sum_{k}^{t+1} = \frac{1}{n \pi_k^{t+1}} p_{ik}^t (x_i - u_k^{t+1})'(x_i - \mu_k^{t+1}) \tag{A4}$$

where $n$ is the number of observations. The algorithm repeats the E- and M-steps iteratively until termination when model parameters converge and the maximum likelihood is reached (convergence of the log-likelihood function) or after a maximum number of iterations.

## Appendix B: Model selection with the BIC and covariance matrix

The Bayesian Information Criterion (BIC) is a criterion for model selection that helps to prevent overfitting by introducing penalty terms for the complexity of the model (number of parameters). In the calculation of the BIC, these penalty terms compete with the likelihood function which determines whether adding parameters improves the model by better fitting the





observed data. In our case, minimizing the BIC achieves a good compromise between keeping the model simple and a good representation of the observed data.

$$BIC(K) = p\,log(n) - 2\,log(L) \tag{B1}$$

460    where $K$ is the number of clusters, $L$ the likelihood of the parameterized mixture model, $p$ the number of parameters to estimate, and $n$ the size of the sample (e.g. 14235 days over 1979-2017). An additional constraint on the definition of clusters is on the covariance matrix. Our GMM is univariate (since we only use PC1) so the variance can be equal or different between clusters (i.e. constraint on volume but not on shape or orientation of clusters).

**Appendix C: Detrended-data based seasonal weather regimes**

465    We first calculate the trend of the Z500 (and TAS) spatial mean over the whole region for each calendar day, and then remove this trend in each grid-cell. A nonlinear cubic smoothing spline is applied for the detrending in order to capture most of the trends correctly (not shown). However, doing only this would result in losing the seasons. Therefore, after removing the trend, we add the estimated seasonal cycle of 2017 as the reference cycle in order to keep a stationary seasonality.