# Peer review of "Seasonal weather regimes in the North Atlantic region: towards new seasonality?"

_Earth System Dynamics, 2020_

## Referee Comment (RC1) · Anonymous Referee #1 · 20 Jul 2020

Overall comments:

This study aims to investigate changes in the weather seasonality in the North Atlantic. Seasonal weather regimes (SWRs) are defined using cluster analysis based on the first principal component on raw Z500 data. Results for four and seven regimes are presented for ERA-Interim and 12 CMIP5 models. Models results are first compared to ERA-Interim for the period 1979-2017. The authors investigate changes in the patterns and frequency of the regimes by comparing CMIP5 simulations for present (1980-2008) and future (2071-2100) climate.

The paper includes a lot of Figures (10 in the main document and 22 in the supplement), with little explanation and description. The authors compare SWRs to the classical weather regimes, but no apparent link exists between the two concepts. The

seasonality analysis is based on daily data but it is not related to weather phenomena (i.e. during winter only 1 regime is found).

Specific comments:

Major:

1.  Definition of seasonal weather regimes (SWRs) and comparison to classical "weather regime". I have trouble to see the link between the author's definition of SWRs and the classical weather regimes. The authors compute the first EOF of raw Z500, which should represent the seasonal cycle (no Figure of EOF1 is provided). SWRs are then defined by clustering PC1 (a single 1D variable). So these clusters should represent the strength of the seasonal cycle, which can be seen for example in Figure 2 by the Z500 contours (stronger Z500 gradient in winter R1, weaker in summer R4). It is also clear from Figure 1 that during winter only R1 occurs. Does it mean that there is no weather in winter and all days look like R1 (cf. for example with Figure 2 in Michel and Riviere 2011)? I find therefore misleading to match the patterns in Figure 2 to the classical weather regimes (see also comment on Figure 2), as those methods are considering two different things. To calculate classical weather regimes a cluster analysis is applied to more than 10 PCs (that explain at least 80% of the variance) and the seasonal cycle is removed from the raw data (e.g. Cassou 2008, Vrac and You 2010). Thus, the mix of different concepts makes it hard to understand what is the main goal of the paper and what the authors attempt with their analysis. a) If the goal is to explain changes in the seasonality by analyzing PC1 of raw Z500 (what it seems so far), I would recommend comparing EOF1 patterns and the distribution of PC1. Would this be a different way of investigating seasonality as compared to previous studies (stated in the Intro, l. 25, without many details)? b) If the goal is to understand future changes in seasonality by looking at the changes in the frequency of weather regimes, those regimes should be defined accordingly (see for example Cattiaux et al. 2013 or Grams et al 2017). However, this task might be challenging, as most climate models have a large bias in weather phenomena (such as blocking, e.g. Masato et al 2013).

2. I miss several explanations in the data and methodology. For example, figure S5 shows seasonal anomalies of TAS, but I could not find how these are defined (neither in the methods, l. 75 nor in the text l. 153). Since one of the main points of the paper should be about changes in the season, the definition of seasonal anomalies needs to be clarified. The same is true for Z500 anomalies.

3. The authors argue for 4 and 7 regimes based on the BIC (without showing it). However, it is not clear what is the main advantage of using 7 regimes. With 4 regimes, R1-4 patterns for 1979-2008 and 2071-2010 are still similar (Fig. S6), while this is not the case with 7. For example, R7 in 1979-2008 (Fig. 3) represents 54 days (0%) and it is argued that this regime becomes more frequent in the future (24%), but R7 for 2071-2010 is very different from R7 in 1979-2008. My recommendation is to present a complete analysis for either 4 or 7 regimes, but not jumping back and forth (i.e. some Figures for 4 are in the supplement, some in the main text).

Specific comments:

Title: I find the word "weather" in the title non-appropriate and confusing. It is strange to call "weather" something that persists for one winter/season. Circulation regimes or seasonal regimes (without weather) might be a better option.

l. 32 "North Atlantic atmospheric patterns are the results of two systems operating at different scales". Please rephrase this paragraph, adding the relevant Literature. A few critical points: low-frequency structures are not "systems". What is the "atmospheric dynamic variability"?

l . 34 anticyclones are also important.

l. 42 How can climate dynamics have a strong seasonal feature? The climate in the extratropics has a strong seasonality. Should atmospheric blocking (time scales of 1-2 weeks) be an example of climate dynamics?

l. 30-40: There is missing relevant literature (for example, there is a bulk of literature

on the eddy-driven jet more relevant than Cassou et al 2008, l. 35). See also some suggestions that could be useful at the end of the document.

l. 49-51 what is the link between the representation of the atmospheric circulation and the seasonality in the model? I agree that it is important that both are correctly represented, but I can not see the link between the two concepts yet. Are the authors suggesting for example, that if the jet stream moves too slowly polewards through the season, this will have an impact on seasonality?

l. 83-84 I expect the first EOF of raw data to be the seasonal cycle. Is that correct?

l. 84 What is the main advantage of a GMM if only a single PC is used for clustering? How different is this method from dividing PC1 into quantiles? Is the PDF of PC1 non-gaussian?

l.104: Instead of only adding the formula in the appendix, it would be very useful to have a figure showing the BIC for the different k in the appendix/supplement

l.112: why adding the seasonal cycle of 2017 and not the seasonal cycle of 1979-2017? What is the reason beyond this choice? This is particularly relevant if the same is done for temperatures.

l. 130 "They are also visually similar to the usual North-Atlantic weather regimes from the literature (e.g. Cassou (2008), Yiou and Nogaj (2004))." I do not think that this is true. Weather regimes are defined by removing the seasonal cycle and mostly using only winter months, why here the "regimes" represent the seasonal cycle. I can not see any Atlantic ridge, or blocking regime here! R3 does not have a ridge over the Atlantic (see comments below Figure 2).

l.162 What is increasing 90hPa? Z500 should be in m or gpm

l. 165 "GFDL-CM3 stands out from the other GCMs by showing the emergence in the future of a new summer regime that did not exist in the past." I see actually a discrepancy between the GFDL-CM3 model for 1979-2008 (figure S8, solid line) and

1979-2017 (Figure 1), so I have trouble to understand this statement.

l. 173-177 It is not clear which trends are meant here and how they are calculated. I do not understand how these trends are calculated since the regimes are not continuous. More explanation is needed here.

l.183-184 Are the regimes (EOF and GMM) calculated for 1979-2100 and then separated in 1979-2008 and 2071-2100? If so I would expect to see a change in the frequency, but not a change in the patterns. For the 7 regimes, I do not see a good correspondence between the patterns (e.g. R2 and R7 in Fig 3 and 4, both shading and contours). Also, I expect the period 2071-2010 to be warmer than 1979-2008, but there are no regimes with warm TAS. This might be linked to how TAS anomalies are defined (please see main comment 2).

l. 185 I see that R7 occurs in summer, but I do not see from any Figures that this is linked to blocking over Scandinavia (Z500 anomalies are over most of the North Atlantic). Also, the percentage (54 days or 0%) is very low compared to blocking frequency (see for example Figures 2 and 3 in Pfahl et al 2012). Moreover, why is R7 much more frequent in the future, but not showing any temperature anomalies? Heatwaves are expected to be linked to blocking also in the future (see e.g. Schaller et al 2018).

l. 218-220: I am not sure to understand the logic behind detrending the data and then calculate the trend of the detrended data. I would understand to i) detrend the data to compute TAS and Z500 anomalies and then compare these for 1979-2008 and 2071-2100 or ii) detrend the data to compute the regimes and look at the trends in the regime occurrence (e.g. trends of Figure 6 and 7), but I do not understand why computing the trends of the detrended anomalies.

l. 220: I can not see the disagreement between the models in Fig. S17

l. 225-235: As I do not understand what has been done to "examine the regime spatial

trends with LGI but without the seasonal shift" I can not comment on this part. Also, what are the "large-scale increases in Z500"?

l. 240-245: Which Figure leads to this conclusion? Why are ERAI and CMIP5 models similar in Figure 1, but very different in the supplement (Figure S8, 1979-2008, solid lines). Also, what is the "increasing frequencies of historical summer conditions of atmospheric dynamics"?

l. 248-270 Which Figures are showing that? Please, add a reference to help to follow the train of thoughts. (I.e. where are cold spells and heatwaves in the analysis? I see only temperature anomalies of a few degrees)? Also, how can a regime be replaced?

Appendix C: Why not having everything in the section "Seasonal weather regimes based on detrended data". Is the same trend removed at each grid cell? Calculated over which region?

Figures:

I think it would very useful to show EOF1 and the explained variance for ERA-Interim and CMIP5 and compare them, before starting calculating and comparing regimes.

Figure 2: I do not see any resemblance of R1-4 with the weather regimes (e.g. from Cassou 2008, https://www.nature.com/articles/nature07286, Figure 1), and I would not expect to see any.

Figure 10: Why are TAS patterns opposite over land and ocean in 1979-2008 and 2071-2010? Is this because the same trend (if I understood it correctly) is removed at each grid point? I expect trends over land and over ocean to be very different (see e.g. Hegerl et al. 2018, Figure 1). Why we do not see this behaviour in Z500? Are the trends in TAS much larger than Z500?

Supplementary Figure 18: Why is the sum of the detrended trends not zero? I would expect some regimes to have positive trends and other regions to have negative trends at the same grid point. Since it is not clear how these trends are calculated, it is difficult

to interpret these Figures.

References

Cattiaux, J., Douville, H., and Peings, Y. (2013). European temperatures in CMIP5: origins of present-day biases and future uncertainties. Climate dynamics, 41(11-12), 2889-2907.

Grams, C. M., Beerli, R., Pfenninger, S., Staffell, I., and Wernli, H. (2017). Balancing Europe's wind-power output through spatial deployment informed by weather regimes. Nature Climate Change, 7(8), 557-562.

Hegerl, G. C., Brönnimann, S., Schurer, A., and Cowan, T. (2018). The early 20th century warming: anomalies, causes, and consequences. Wiley Interdisciplinary Reviews: Climate Change, 9(4), e522.

Lorenz, D. J., and Hartmann, D. L. (2003). Eddy–zonal flow feedback in the Northern Hemisphere winter. Journal of climate, 16(8), 1212-1227.

Masato, G., Hoskins, B. J., and Woollings, T. (2013). Winter and summer Northern Hemisphere blocking in CMIP5 models. Journal of Climate, 26(18), 7044-7059.

Michel, C., and Rivière, G. (2011). "The link between Rossby wave breakings and weather regime transitions." Journal of the Atmospheric Sciences 68.8: 1730-1748.

Pfahl, S., and Wernli, H. (2012). Quantifying the relevance of atmospheric blocking for co-located temperature extremes in the Northern Hemisphere on (sub-) daily time scales. Geophysical Research Letters, 39(12).
* * *

---

## Referee Comment (RC2) · Anonymous Referee #2 · 6 Aug 2020

The authors utilize the concept of seasonal weather regimes defined based on clustering daily fields of 500 hPa geopotential height from reanalysis and model data. By exploring their frequency and distribution throughout the year, they are able to trace changes in seasonality of European/North Atlantic climate, which are further explored in the course of the study. In addition to the material presented in the main text, the supplementary material contains an impressive body of additional figures illustrating the results of this study in more detail and may be helpful to better understand certain aspects of the authors' findings. The manuscript itself is concise, well-written and contains (almost) all necessary information needed to fully comprehend the presented analysis. I only have a few minor comments listed below that I would like to invite the authors to consider/address before this work may become published as a regular paper

in ESD.

My main point concerns some minor clarification on the use of the EM algorithm for clustering the daily Z500 fields. Specifically, I have two questions:

1. Could you please elaborate a bit more explicitly on the meaning of the variable x that you use for clustering? Is it really just the (scalar) PC 1 score (i.e. you collapse the full spatial pattern to the magnitude of the EOF-1 pattern), and you perform a one-dimensional cluster analysis?

2. I would appreciate if you could motivate a bit more explicitly your use of a model-based clustering technique as opposed to other model-free approaches like hierarchical or k-means clustering. How is this related to the choice/type of the variable x that the clustering is based on? Why is a finite Gaussian mixture a reasonable model for the designated purpose? I think that the obtained results are correct and reasonable, I just want to better understand the rationale behind the approach followed here.

Further minor points:

3. I found the description of the detrending procedure a bit hard to follow in the main text. I understand from Appendix C that you take the daily mean of the whole Z500 field and subtract it from all values (thereby removing not only long-term trends, but also seasonality in the regional-mean Z500 amplitudes)? Or is it something different? I have to confess that I am a bit lost with the term "calendar trend" (ll.111 and 215).

4. There are quite a few cases where additional technical results are "not shown". Since you provide a very detailed supplementary material with many additional figures, I was wondering if it would make sense to also include (some of) the results labelled as "not shown" in this supplement?

5. In Figure 3.1, I would find it more logical to start with showing and discussing the obtained SWRs (i.e., the associated spatial patterns) before focusing on their past changes.

6. Relating to ll.139-147: In order to better understand the (dis)agreement between the Z500 fields and SWR distributions obtained from them, some quantitative metrics (like mean spatial correlation, bias,. . .) between the models and ERA-Interim could be added as a table to the supplementary material.

7. L.165: Can you elaborate a bit more on the "new summer regime that did not exist in the past"?

8. The authors consider future changes in SWPs over the European/North Atlantic sector for the RCP8.5 scenario only. Did they check if the corresponding results for more moderate scenarios would be compatible with the reported findings (i.e. show consistent trends but smaller "magnitude" of changes)? I do not request to show any additional results for RCPs, but a brief discussion (e.g. along with what is stated in ll.308-309) could be interesting.

Technical suggestions:

* L.2: "insight into"

* L.3: What do you mean by "for human and natural systems"? This reads a bit odd to me. . .

* L.125: "the regimes' monthly frequencies"

* L.176: "regression coefficients"

* L.186: "the northern continents" – please be a bit more specific, as you do not seem to consider the whole hemisphere

* Ll.189-190: "the R1 pattern. . . and the R7 pattern"

* L.195: "the regimes' spatial patterns"

* L.225: I don't quite get what "LGI" stands for – please explain

* L.227: "the regimes' spatial trends" (what is a "spatial trend"?)

\* L.449, Eq. (A4): Please replace $\sum_{k}^{t+1}$ by $\Sigma_k^{t+1}$ in the LATEX source, the presently shown mathematical symbol is not appropriate here

---

## Author Comment (AC1) · 30 Sep 2020

**Response to Referee Comment 1 on "Seasonal weather regimes in the North Atlantic region: towards new seasonality?" by Florentin Breton et al.**

**Comment**

This study aims to investigate changes in the weather seasonality in the North Atlantic. Seasonal weather regimes (SWRs) are defined using cluster analysis based on the first principal component on raw Z500 data. Results for four and seven regimes are presented for ERA-Interim and 12 CMIP5 models. Models results are first compared to ERA-Interim for the period 1979-2017. The authors investigate changes in the patterns and frequency of the regimes by comparing CMIP5 simulations for present (1980-2008) and future (2071-2100) climate. The paper includes a lot of Figures (10 in the main document and 22 in the supplement), with little explanation and description. The authors compare SWRs to the classical weather regimes, but no apparent link exists between the two concepts. The seasonality analysis is based on daily data but it is not related to weather phenomena (i.e. during winter only 1 regime is found).

**Response:**

We thank the referee for the comments. However, some of these comments reflect misunderstandings of the paper which we will address in our response. We will update the manuscript with the suggestions and clarifications in relation to the comments. Our response to each comment can be found below.

**Comment**

1. Definition of seasonal weather regimes (SWRs) and comparison to classical "weather regime". I have trouble to see the link between the author's definition of SWRs and the classical weather regimes. The authors compute the first EOF of raw 2500, which should represent the seasonal cycle (no Figure of EOF1 is provided). SWRs are then defined by clustering PC1 (a single 1D variable). So these clusters should represent the strength of the seasonal cycle, which can be seen for example in Figure 2 by the Z500 contours (stronger Z500 gradient in winter R1, weaker in summer R4). It is also clear from Figure 1 that during winter only R1 occurs. Does it mean that there is no weather in winter and all days look like R1 (cf. for example with Figure 2 in Michel and Riviere (2011)? I find therefore misleading to match the patterns in Figure 2 to the classical weather regimes (see also comment on Figure 2), as those methods are considering two different things. To calculate classical weather regimes a cluster analysis is applied to more than 10 PCs (that explain at least 80% of the variance) and the seasonal cycle is removed from the raw data (e.g. Cassou 2008, Vrac and You 2010). Thus, the mix of different concepts makes it hard

to understand what is the main goal of the paper and what the authors attempt with their analysis.

a) If the goal is to explain changes in the seasonality by analyzing PC1 of raw Z500 (what it seems so far), I would recommend comparing EOF1 patterns and the distribution of PC1. Would this be a different way of investigating seasonality as compared to previous studies (stated in the Intro, I. 25, without many details)?
b) If the goal is to understand future changes in seasonality by looking at the changes in the frequency of weather regimes, those regimes should be defined accordingly (see for example Cattiaux et al. (2013) or Grams et al. (2017)). However, this task might be challenging, as most climate models have a large bias in weather phenomena (such as blocking, e.g. Masato et al 2013).

**Response:**

The link between our SWRs and the classical weather regimes is the analogy in the general way they are defined (i.e. clustering approach by classifying similar atmospheric situations). However, our preprocessing of the data and their constraints to define the regimes are different, and thus their temporal properties are also different, compared to classical weather regimes. Indeed, our goal is not to study the intra-seasonal variability with classical circulation regimes but rather to study the variability of the seasonal cycle via clustering-based regimes of circulations. Some of the seasonal structures that we find happen to be similar to some of the intra-seasonal structures. A detailed explanation is provided below.

Our SWRs are defined in the same way as in the original paper (Vrac et al. 2014) that investigated past changes in the seasonality of daily atmospheric circulation (here we compare historical simulations to a reanalysis, before investigating future simulations). So, they are defined similarly to "classical" weather regimes except that we take full-year data (instead of summer or winter only; lines 76-77) and raw values (instead of seasonal anomalies; lines 76-77). Indeed, we are studying how the seasonal structures might migrate in the calendar year (e.g. from summer towards winter or vice-versa). Our regimes are "seasonal" because they are defined (and studied) based on their seasonality (weather patterns and annual cycle). We compare them to classical weather regimes to highlight weather patterns that appear similar (lines 129-137), while remaining cautious about the differences in their definition and properties (lines 137-138). It is important to keep in mind that the SWRs are defined based on the daily PC1 values of Z500 and that the figures 1 and 2 represent averages of the atmospheric conditions (respectively the frequency of occurrence and the spatial patterns) for the days belonging to the SWRs over the period. Indeed, the days are attributed to the regimes based on their similarity in terms of atmospheric situations, and there are also many days of R2 in winter (not only R1), or R3 in summer (not only R4). Since our main goal is to study the response of the atmospheric circulation seasonality to climate change, this attribution is based rather on the long-term variability of seasonality rather than intra-seasonal variability.

Regarding PC1, it contains not only most of the seasonal cycle, but also a large part of the long-term variability, as shown on Figure A below.

Figure A. Spectral power captured by each PC in function of the period. Dataset: Z500 from ERAI over 1979-2017.

Furthermore, the pattern given by the first EOF and the distribution of PC1 for ERAI and climate models appear generally similar, as shown respectively on Figures B and C below. The pdf of PC1 approximately corresponds to a bimodal Gaussian-like distribution (Fig. C).

---

## Author Comment (AC2) · 30 Sep 2020

**Response to Referee Comment 2* on **"Seasonal weather regimes in the North Atlantic region: towards new seasonality?"** *by* **Florentin Breton et al.**

**Comment:**
*The authors utilize the concept of seasonal weather regimes defined based on clustering daily fields of 500 hPa geopotential height from reanalysis and model data. By exploring their frequency and distribution throughout the year, they are able to trace changes in seasonality of European/North Atlantic climate, which are further explored in the course of the study. In addition to the material presented in the main text, the supplementary material contains an impressive body of additional figures illustrating the results of this study in more detail and may be helpful to better understand certain aspects of the authors' findings. The manuscript itself is concise, well-written and contains (almost) all necessary information needed to fully comprehend the presented analysis. I only have a few minor comments listed below that I would like to invite the authors to consider/address before this work may become published as a regular paper in ESD.*

**Response:**
We thank the referee for the positive, constructive and helpful comments and detailed remarks. We will update the manuscript with the suggestions and clarifications in relation to the comments. Our response to each comment can be found below.

**Comment:**
*My main point concerns some minor clarification on the use of the EM algorithm for clustering the daily Z500 fields. Specifically, I have two questions:*
*1. Could you please elaborate a bit more explicitly on the meaning of the variable x that you use for clustering? Is it really just the (scalar) PC1 score (i.e. you collapse the full spatial pattern to the magnitude of the EOF-1 pattern), and you perform a one-dimensional cluster analysis?*

**Response:**
Yes, we use only PC1 values in our clustering because our focus is on seasonality which is overwhelmingly captured by PC1 (see lines 82-88). In the preliminary steps of the analysis, we tested the sensitivity of the results to the number of PCs included in the clustering. In most cases, including a few more PCs (e.g. from 2 to 5 PCs) brings similar results (weather patterns, seasonal cycle). Therefore, we use only PC1 and the cluster analysis is indeed one-dimensional (see lines 92-93). We propose the following revisions (blue) in the manuscript (line 108):

"*We also tested the sensitivity of the clustering results (weather patterns, annual cycle) to the number of PCs included (from 1 to 5). There was a small influence of additional PCs on the results (reanalyses, models) over 1979-2017 and very small influence over 1979-2100 (increasing with the number of PCs; not shown). This reinforced our choice of using only the first PC, considering that additional PCs represent little additional seasonality and difference in the long-term response of atmospheric circulation to climate change.*"

**Comment:**

*2. I would appreciate it if you could motivate a bit more explicitly your use of a model-based clustering technique as opposed to other model-free approaches like hierarchical or k-means clustering. How is this related to the choice/type of the variable x that the clustering is based on? Why is a finite Gaussian mixture a reasonable model for the designated purpose? I think that the obtained results are correct and reasonable, I just want to better understand the rationale behind the approach followed here.*

**Response:**

The choice of a finite Gaussian mixture is justified by the distribution of PC1 (x) that is Gaussian-like and bimodal (seasons), as shown on the figure below.

[Figure]

K-means is a distance-based clustering method that assumes spherical clusters and is more sensitive to noise (Estivill-Castron & Yang, 2000; Han et al., 2011; Rokach & Maimon, 2005). EM is a probabilistic model-based clustering method that can be seen as more statistically sophisticated than k-means, although it requires more computation and detailed prior knowledge (Estivill-Castron & Yang, 2000). The EM approach generalizes k-means in allowing more flexibility in the covariance structure

of the clusters (Banfield and Raftery, 1993; Rust et al., 2010): k-means corresponds to a special case of EM with spherical Gaussian laws that is less able to capture non-spherical structures. Hierarchical clustering was considered less suitable than EM because of the inability to scale well with a large number of values (about 45000 for PC1 over 1979-2100; Rokach & Maimon, 2005) and with unsupervised decisions of merging or splitting without examining or evaluating many objects or clusters (Han et al., 2011). In the case of EM using only PC1 (univariate data), the initial partitioning of data is done by separating in quantiles, so it is less sensitive to noise, outliers and extremes. We tried both EM and k-means algorithms in the preliminary stage of our study to test the sensitivity of the results to the clustering method, they brought similar results so we focused on EM. We propose the following revisions (blue) in the manuscript (lines 107-108):

> *"Different clustering methods can lead to different results (e.g., Philipp et al. (2010)) so we tested the sensitivity of the SWR results to using the k-means clustering algorithm (more popular but less flexible; Estivill-Castro and Yang, 2000, Han et al., 2011, Lior and Maimon, 2005) instead of EM, which brought very similar results (not shown). EM can be seen as a generalization of k-means with less constraint on the shape of clusters and better ability to account for structures of arbitrary shape (Han et al., 2011, Lior and Maimon, 2005)."*

**Comment:**
*3. I found the description of the detrending procedure a bit hard to follow in the maintext. I understand from Appendix C that you take the daily mean of the whole Z500 field and subtract it from all values (thereby removing not only long-term trends, but also seasonality in the regional-mean Z500 amplitudes)? Or is it something different? I have to confess that I am a bit lost with the term "calendar trend" (ll.111 and 215).*

**Response:**
Yes, to clarify the description of the detrending procedure, we propose the following revisions (blue) in the manuscript (lines 110-112) and the removal of "Appendix C":

> *"The goal now is to remove the large-scale increase of Z500 to further investigate changes in Z500 patterns. This requires to preserve both the spatial structures and the seasonality while removing the large-scale effect. Calculating and removing the trend by gridpoint would result in losing the spatial structures while doing so without a year of reference would result in losing the seasons. Therefore, the trend is calculated on the spatial mean of the whole area for each calendar day, with reference to 2017. This means that for each specific day of the calendar year (January 1st only, ..., December 31st only) the trend is calculated with the 122 values (from 1979 to 2100) of the spatial mean for this specific day. We took 2017 as the reference year because it is the last year contained in both*

*reanalyses and models. The trend was estimated best by using a cubic smoothing spline.*"

**Comment:**
*4. There are quite a few cases where additional technical results are "not shown". Since you provide a very detailed supplementary material with many additional figures, I was wondering if it would make sense to also include (some of) the results labelled as "not shown" in this supplement?*

**Response:**
We considered at length which figures to show in the paper (and supplement) and chose to remove those that show very similar results or bring little more information to the analysis, trying to find the balance between making the research investigation understandable and keeping the paper readable (not too long).

**Comment:**
*5. In Figure 3.1, I would find it more logical to start with showing and discussing the obtained SWRs (i.e., the associated spatial patterns) before focusing on their past changes.*

**Response:**
If we understand correctly, the comment from the reviewer suggests to show and discuss the spatial patterns of SWRs before looking at their annual cycle in Section 3.1. We agree with the comment and propose the following revisions (blue) in the manuscript (lines 125-138 and Figures 1-2):

*We start by looking at the weather patterns of the regimes as shown by the composite maps in Figure 1. For climate models, each regime composite map is determined individually (i.e. average map) and the multimodel composite is calculated as the mean of the distribution of the twelve composites. The spatial patterns of the four average regimes found in the models are very similar to those from ERAI. They are also visually similar to the usual North-Atlantic weather regimes from the literature (e.g. Cassou (2008), Yiou and Nogaj (2004)).The first regime (R1) corresponds to the positive phase of the North Atlantic Oscillation (NAO+) and the second (R2) to its negative phase (NAO-; Hurrell et al. (2003)). The third and fourth regimes (R3 and R4) respectively represent the AtlanticRidge (AR) and Scandinavian Blocking (SB) atmospheric conditions, resembling the weather patterns from Yiou and Nogaj (2004), and Vrac et al. (2014). However, note that the temporal patterns of our SWRs are based on full years (like Vrac et al. (2014)), unlike the literature considering weather patterns in winter (Cassou (2008), Yiou and Nogaj (2004)) or in summer(e.g. Guemas et al. (2010)). Thus, if our SWRs resemble the usual regimes, they present differences in their definition and properties. The*

*annual cycle of our regimes' monthly frequencies over 1979-2017 is shown in Figure 2. Climate models reproduce a seasonal cycle of SWRs similar to ERAI, with regime 1 (hereafter R1) representing a winter-like season,R4 a summer-like season, and R2 and R3 transitional seasons (R2 around winter and R3 around summer).*

[Figure]

**Figure 1.** Composite maps of the four regimes (one per row) for ERAI (first column) and climate models (second column; each map shows the average of 12 composite maps; third column shows standard deviation of Z500 between the 12 composites). The maps are calculated by averaging the seasonal anomalies (shading) and raw values (contour lines) over the days belonging to the regime. Seasonal anomalies correspond to the raw values minus the average seasonal cycle. The number of days per regime is shown above each map (average of 12 values for the climate models).

[Figure]

**Figure 2.** Average seasonal cycle of the frequencies of occurrences for the four regimes of ERAI and the 12 climate models, over 1979-2017. Monthly frequencies correspond to the number of days of regime occurrence divided by the number of days in the month.

**Comment:**

*6. Relating to ll.139-147: In order to better understand the (dis)agreement between the Z500 fields and SWR distributions obtained from them, some quantitative metrics(like mean spatial correlation, bias,...) between the models and ERA-Interim could be added as a table to the supplementary material.*

**Response:**

Thank you for this interesting idea. We calculated the coefficients of pattern correlation between the regimes from the different models and ERAI, and propose to include these results in Table 2 of the main materials of the paper (with the following changes highlighted in blue; lines 139-140):

*"In general, the climate models reproduce atmospheric weather patterns that are very similar to ERAI, but individual models are less successful (see Table 2 and Supplementary Fig. 1-4)."*

**Table 2.** Coefficients of pattern correlation between the regimes from ERAI and each climate model over 1979-2017.

| Model | R1 | R2 | R3 | R4 |
|---|---|---|---|---|
| Average | 0.901 | 0.869 | 0.518 | 0.9 |
| ACCESS1-3 | 0.944 | 0.888 | 0.275 | 0.805 |
| bcc-csm1-1-m | 0.928 | 0.842 | 0.757 | 0.959 |
| CanESM2 | 0.935 | 0.888 | 0.781 | 0.931 |
| CNRM-CM5 | 0.937 | 0.933 | 0.589 | 0.962 |
| GFDL-CM3 | 0.934 | 0.868 | 0.486 | 0.932 |
| HadGEM2-ES | 0.926 | 0.879 | 0.887 | 0.916 |
| IPSL-CM5A-MR | 0.908 | 0.874 | 0.529 | 0.948 |
| IPSL-CM5B-LR | 0.912 | 0.859 | 0.65 | 0.933 |
| MIROC5 | 0.553 | 0.674 | -0.397 | 0.824 |
| MPI-ESM-MR | 0.968 | 0.934 | 0.59 | 0.837 |
| MRI-ESM1 | 0.913 | 0.86 | 0.753 | 0.815 |
| NorESM1-M | 0.95 | 0.925 | 0.319 | 0.941 |

**Comment:**

*7. L.165: Can you elaborate a bit more on the "new summer regime that did not exist in the past"?*

**Response:**

The main finding expressed here (and developed further in the paper) is the emergence of a new regime of atmospheric circulation that was not present in the historical period. As the clustering has little freedom with 4 clusters, this emergence is more evident with 7 clusters (as shown in the later results). An interesting point is that GFDL is the only model showing the emergence of a new regime with only 4 clusters (i.e. despite large constraints on the definition of the regimes). It means that in the case of GFDL, the difference between historical and future Z500 conditions in summer is so large that a new regime was created by the clustering algorithm. However, future R4 appears relatively similar between models (see Figure 4 below from the Supplementary materials of the paper):

[Figure]

Since future R4 in GFDL is very similar to future R4 from other models (for which R4 was already well established in the past), the emergence of R4 in GFDL doesn't represent the emergence of a new regime from a general perspective. We propose the following changes (highlighted in blue) in the manuscript (lines 164-165):

*"GFDL-CM3 stands out from the other GCMs by showing the emergence in the future of a new summer regime that almost did not exist in the past (one day, not shown). This emergence means that in the case of GFDL, the difference between historical and future Z500 conditions in summer is so large that a new regime was created in the clustering. As the clustering has little freedom with 4 clusters (i.e. large constraints on the definition of the regimes), this emergence is even more interesting, but it is consistent with higher increases of Z500 and TAS in this climate model by comparison to other models (not shown). However, since future R4 in GFDL is very similar to future R4 from other models (seasonal cycle and weather pattern, respectively shown in Supplementary Fig. 8 and not shown), and since R4 was already well established in the past for other models, this emergence of R4 in GFDL does not represent the emergence of a new regime from a general perspective."*

**Comment:**

*8. The authors consider future changes in SWRs over the European/North Atlantic sector for the RCP8.5 scenario only. Did they check if the corresponding results for more moderate scenarios would be compatible with the reported findings (i.e. show consistent trends but smaller "magnitude" of changes)? I do not request to show any additional results for RCPs, but a brief discussion (e.g. along with what is stated in ll.308-309) could be interesting.*

**Response:**

RCP8.5 was initially chosen because it contains the largest signal of anthropogenic climate change and therefore facilitates the emergence of long-term changes in the climate system (as in many climate studies). We decided not to try other scenarios because it was expected (as said by the reviewer) to find similar trends of smaller magnitude to RCP8.5 (increasing in proportion to the scenario forcing), and the later emergence if any of the new regime in summer (increasing earliness in proportion to the scenario forcing). As the current trajectory of the Earth system already approximately follows the RCP8.5 scenario so far (Allen et al., 2018), the choice of scenarios representing its future trajectory is debated (Burgess et al., 2020; Hausfather & Peters, 2020), but RCP8.5 appears to remain a solid figure for the decades to come (Schwalm et al., 2020).

We propose the following changes (highlighted in blue) in the manuscript (line 80):

"*The choice of the RCP8.5 scenario is motivated by its approximative representation of the current climate trajectory (Allen et al., 2020), its plausibility for future climate trajectory (Schwalm et al., 2020), and its large magnitude of scenario forcing for facilitating the emergence of long-term changes in the climate system.*"

**Comment:**
*\* L.2: "insight into"*

**Response:**

Thank you, we revised the manuscript accordingly (line 2).
"*As both have strong seasonal features, a better insight into their future seasonality is essential to anticipate changes in weather conditions for human and natural systems.*"

**Comment:**
*\* L.3: What do you mean by "for human and natural systems"? This reads a bit odd to me...*

**Response:**

We are differentiating between the impacts on nature (e.g. phenology of ecosystems) and society (e.g. health, transportation, energy). We propose the following changes (highlighted in blue) in the manuscript to make it clearer (lines 3 and 28-29).

*"As both have strong seasonal features, a better insight into their future seasonality is essential to anticipate changes in weather conditions for human systems (e.g. health, transportation, energy) and natural systems (e.g. phenology)."*

*"The meteorological seasons are a prominent feature of climate variability, experienced by human systems (e.g. health, transportation, energy) and natural systems (e.g. phenology) through the seasonality of surface weather conditions."*

**Comment:**
*\* L.125: "the regimes' monthly frequencies"*

**Response:**
Thank you, we revised the manuscript accordingly (line 125).
*"We start by looking at the annual cycle of the regimes' monthly frequencies over 1979-2017, shown in Figure 1."*

**Comment:**
*\* L.176: "regression coefficients"*

**Response:**
Thank you, we revised the manuscript accordingly (line 176).

*"Both regression coefficients and p-values are calculated individually by climate model, and then averaged over the twelve values."*

**Comment:**
*\* L.186: "the northern continents" – please be a bit more specific, as you do not seem to consider the whole hemisphere*

**Response:**
Yes, we revised the manuscript accordingly (line 185-186).

*"Past (1979-2008) R7 corresponds to rare and very intense conditions of Scandinavian Blocking associated with summer heatwaves over the continents of the North Atlantic region (except North Africa and northernmost Canada)."*

**Comment:**
*\* Ll.189-190: "the R1 pattern...and the R7 pattern"*

**Response:**
Thank you, we revised the manuscript accordingly (lines 188-191).

*"Overall, we observe a shift in the spatial patterns (Z500 and TAS) of the regimes (Figures 3-4) with past R1 patterns becoming future R2 patterns, past R2 patterns becoming future R3 patterns, and so on until R6, while the R1 pattern becomes seasonally more extreme (rarer and more intense pattern) and the R7 pattern becomes seasonally more normal (more frequent and less intense pattern)."*

**Comment:**
*\* L.195: "the regimes' spatial patterns"*

**Response:**
Thank you, we revised the manuscript accordingly (lines 188-191).

*"This shift in the seasonal cycle of the regimes between past and future appears very consistent with the shift in the regimes' spatial patterns."*

**Comment:**
*\* L.225: I don't quite get what "LGI" stands for – please explain*

**Response:**
We just noticed that "LSI" should be used to designate "large-scale increase" instead of "LGI" and propose to replace the term in the manuscript. LSI stands for the "large-scale increase" of TAS and Z500 driven by human influence (lines 58-59; Christidis & Stott, 2015), corresponding for Z500 to the atmospheric thermodynamic response to human-caused global warming.

**Comment:**
*\* L.227: "the regimes' spatial trends" (what is a "spatial trend"?)*

**Response:**
Thank you, we revised the manuscript accordingly. We used the term "spatial trends" to denote the geographic variation of the trends calculated for each gridpoint. To clarify the manuscript, we propose to replace the term by "spatial pattern of trends" or "trends" where suitable.

**Comment:**
*\* L.449, Eq. (A4): Please replace $\sum_{k}^{t+1}$ by $\Sigma_k^{t+1}$ in the LATEXsource, the presently shown mathematical symbol is not appropriate here*

**Response:**
Thank you, we revised the manuscript accordingly (Equation 4).

*"\begin{equation}*

$$\Sigma_k^{t+1} \, = \, \frac{1}{n \, \pi_k^{t+1}} \, p_{ik}^t \, ( \, x_i \, - \, u_k^{t+1} \, )' \, ( \, x_i \, - \, \mu_k^{t+1} \, )$$
\end{equation}"

**References:**

Allen, M.R., O.P. Dube, W. Solecki, F. Aragón-Durand, W. Cramer, S. Humphreys, M. Kainuma, J. Kala, N. Mahowald, Y. Mulugetta, R. Perez, M. Wairiu, and K. Zickfeld, 2018: Framing and Context. In: Global Warming of 1.5°C. An IPCC Special Report on the impacts of global warming of 1.5°C above pre-industrial levels and related global greenhouse gas emission pathways, in the context of strengthening the global response to the threat of climate change, sustainable development, and efforts to eradicate poverty [Masson-Delmotte, V., P. Zhai, H.-O. Pörtner, D. Roberts, J. Skea, P.R. Shukla, A. Pirani, W. Moufouma-Okia, C. Péan, R. Pidcock, S. Connors, J.B.R. Matthews, Y. Chen, X. Zhou, M.I. Gomis, E. Lonnoy, T. Maycock, M. Tignor, and T. Waterfield (eds.)]. In Press

Banfield, J. D., & Raftery, A. E. (1993). Model-based Gaussian and non-Gaussian clustering. *Biometrics*, 803-821.

Burgess, M. G., Ritchie, J., Shapland, J., & Pielke Jr, R. (2020). IPCC baseline scenarios over-project CO2 emissions and economic growth.

Christidis, N., & Stott, P. A. (2015). Changes in the geopotential height at 500 hPa under the influence of external climatic forcings. Geophysical Research Letters, 42(24), 10-798.

Estivill-Castro, V., & Yang, J. (2000, August). Fast and robust general purpose clustering algorithms. In *Pacific Rim International Conference on Artificial Intelligence* (pp. 208-218). Springer, Berlin, Heidelberg.

Han, J., Pei, J., & Kamber, M. (2011). *Data mining: concepts and techniques*. Elsevier.

Hausfather, Z., & Peters, G. P. (2020). Emissions–the 'business as usual' story is misleading.

Rokach L., Maimon O. (2005) Clustering Methods. In: Maimon O., Rokach L. (eds) Data Mining and Knowledge Discovery Handbook. Springer, Boston, MA. https://doi.org/10.1007/0-387-25465-X_15

Rust, H. W., Vrac, M., Lengaigne, M., & Sultan, B. (2010). Quantifying differences in circulation patterns based on probabilistic models: IPCC AR4 multimodel comparison for the North Atlantic. *Journal of Climate*, *23*(24), 6573-6589.

Schwalm, C. R., Glendon, S., & Duffy, P. B. (2020). RCP8. 5 tracks cumulative $CO_2$ emissions. *Proceedings of the National Academy of Sciences*, *117*(33), 19656-19657.

Schwalm, C. R., Glendon, S., & Duffy, P. B. (2020). RCP8. 5 tracks cumulative $CO_2$ emissions. *Proceedings of the National Academy of Sciences*, *117*(33), 19656-19657.

Vrac, M., Vaittinada Ayar, P., & Yiou, P. (2014). Trends and variability of seasonal weather regimes. *International journal of climatology*, *34*(2), 472-480.